# IMPROVING NON-AUTOREGRESSIVE TRANSLATION MODELS WITHOUT DISTILLATION

**Xiao Shi Huang, Felipe Pérez, Maksims Volkovs**
Layer 6 AI
`{gary,felipe,maks}@layer6.ai`

## ABSTRACT

Transformer-based autoregressive (AR) machine translation models have achieved significant performance improvements, nearing human-level accuracy on some languages. The AR framework translates one token at a time which can be time consuming, especially for long sequences. To accelerate inference, recent work has been exploring non-autoregressive (NAR) approaches that translate blocks of tokens in parallel. Despite significant progress, leading NAR models still lag behind their AR counterparts, and only become competitive when trained with distillation. In this paper we investigate possible reasons behind this performance gap, namely, the indistinguishability of tokens, and mismatch between training and inference. We then propose the Conditional Masked Language Model with Correction (CMLMC) that addresses these problems. Empirically, we show that CMLMC achieves state-of-the-art NAR performance when trained on raw data without distillation, and approaches AR performance on multiple datasets. Code for this work is available here: `https://github.com/layer6ai-labs/CMLMC`.

## 1    INTRODUCTION

Neural machine translation (NMT) models based on the Transformer architecture have achieved leading performance (Vaswani et al., 2017; Barrault et al., 2019; Huang et al., 2020). Majority of the proposed approaches are based on the autoregressive (AR) principle, where translation is done one token at a time conditioning on already translated tokens. AR inference scales linearly with the number of tokens and full forward pass through the decoder is required for each translated token. This can be prohibitively expensive for long sequences, particularly as leading models are becoming increasingly larger in size. To mitigate this problem, recent works have explored the non-autoregressive (NAR) approach where subsets of tokens are translated in parallel (Gu et al., 2018; Ghazvininejad et al., 2019; Kasai et al., 2020). NAR models achieve significantly faster inference speed that no longer depends on sequence length. However, despite considerable progress, leading NAR models still require sequence-level knowledge distillation (Kim & Rush, 2016) to achieve competitive accuracy. In practice, a large AR Transformer model trained on the raw data is used as the teacher for distillation (Ghazvininejad et al., 2019). This process is expensive, as every new language pair requires training a new teacher. It is also non-standard, and raises questions to the necessity and the underlying problems solved by distillation (Zhou et al., 2020; Ding et al., 2021).

In this work we focus on one of the leading NAR approaches, the Conditional Masked Language Model (CMLM) (Ghazvininejad et al., 2019). CMLM achieved leading NAR performance on multiple NMT datasets - especially when combined with semi-autoregressive training (Ghazvininejad et al., 2020b) - but only when the model is trained on distilled data. Without distillation, CMLM performance drops significantly below AR benchmarks. The need for distillation indicates that CMLM alone is unable to fully leverage the information available in the raw training data (Ding et al., 2021). Here, we identify two shortcomings of CMLM that, when addressed, significantly improve NAR translation quality and narrow the gap between raw and distilled performance.

First, input token representations in CMLM can become nearly indistinguishable, especially for adjacent positions. In AR models this problem is avoided by a combination of causal masked attention, sequential inference, and learned positional encodings (PEs). However, unmasked attention and simultaneous translation of token blocks in CMLM loses most of the information that distinguishes tokens. This problem is particularly severe during the first inference step, where the input is fully masked. The model thus only relies on learned PEs to distinguish tokens, which is not sufficient. Poor token separation can cause significant translation errors, including the identified phenomenon of token repetition stemming from the related multi-modality problem (Zhou et al., 2020).

Second, there is a misalignment between CMLM's training and inference procedures. During training CMLM is optimized with a masked loss analogous to language model training in popular models such as BERT (Devlin et al., 2019). However, CMLM inference always starts with a fully masked sentence and translates all tokens simultaneously. Iterative refinement is then applied where subsets of low confidence tokens are masked and re-translated at each iteration. During training the model rarely sees a fully masked sentence, and is not trained to self-correct from the initial fully masked translation that can contain significant errors. The misalignment between the two procedures can cause a disconnect, where optimization of the training loss does not transfer to improvements in translation quality.

In this work we propose the **C**onditional **M**asked **L**anguage **M**odel with **C**orrection (CMLMC). Our model builds on the CMLM architecture and addresses the aforementioned problems. We modify the decoder structure by exposing the positional encodings and incorporating causal attention layers to differentiate adjacent tokens. We also propose a novel correction loss that teaches the model how to correct translation mistakes made in early decoding iterations from the fully masked sentence. With these improvements, CMLMC achieves new state-of-the-art undistilled NAR results and approaches AR performance on multiple NMT benchmarks.

## 2 RELATED WORK

Neural machine translation is a sequence to sequence prediction problem where a source sentence $X = (x_1, \ldots, x_m)$ in one language is transformed into a target sentence $Y = (y_1, \ldots, y_n)$ in another language. In AR setting this problem is modelled as: $\arg\max_Y \prod_{i=1}^{n} P(y_i | X, Y_{<i})$, where $Y_{<i} = (y_1, \ldots, y_{i-1})$ so every token is conditioned on all translated tokens before it (Cho et al., 2014). Greedy inference approach is typically taken where tokens are translated left-to-right sequentially, so a sequence of length $n$ requires $n$ forward passes through the model. Transformer-based AR models currently achieve leading accuracy on most NMT benchmarks (Barrault et al., 2019), but as models become larger linear inference can become prohibitively expensive for long sequences.

The NAR approach is proposed to mitigate this problem. Early work NAT (Gu et al., 2018) models the problem as: $\arg\max_Y P(n|X) \prod_{i=1}^{n} P(y_i|X)$, where $P(n|X)$ is the probability over sequence length. Since $P(y_i|X)$ is independent of $Y$, inference can be parallelized and all tokens (or their subsets) can be translated simultaneously. The conditional independence assumption can be overly restrictive and typically does not hold in natural language. To relax it, NAT adds positional attention and full self-attention between all tokens. However, the performance of this model is still considerably worse than the AR benchmarks, which authors attribute to the multi-modality problem (Gu et al., 2018). NAT tackle this problem by applying knowledge distillation (Kim & Rush, 2016) on the training data, where a large AR model is used as a teacher. Distillation significantly improves performance, and has since been widely adopted to reduce the performance gap to AR benchmarks.

Recently, multiple NAR approaches have been proposed that build on the work of NAT (Gu et al., 2018). Flowseq (Ma et al., 2019) introduces latent variables and generative flow to decouple token dependencies in the output. CMLM (Ghazvininejad et al., 2019), the focus of our work, extends masked language loss training (Devlin et al., 2019) to NMT to facilitate more robust handling of target language input initialization and iterative refinement. Several works build on CMLM to improve performance such as incorporating the energy of teacher optimization (Tu et al., 2020), adding raw data prior (Ding et al., 2021), and improving the encoder-decoder architecture (Kasai et al., 2021). DisCo (Kasai et al., 2020) utilizes attention masking instead of token masking to improve both training and inference efficiency. ReorderNAT (Ran et al., 2021) uses an explicit module to re-order the input sequence. GLAT (Qian et al., 2021) compares intermediate predictions with ground truth to dynamically adjust the masking percentage according to translation difficulty. Recent work, including AXE (Ghazvininejad et al., 2020a), Imputer (Saharia et al., 2020), FullyNAT (Gu & Kong, 2021), and OAXE (Du et al., 2021) achieve improved performance with the alignment loss (Libovický & Helcl, 2018) that has been found to be more suitable for one-step NAR tasks (Haviv et al., 2021). This important direction is orthogonal to our focus in this paper, and can be readily incorporated in future work. The closest to our work is the SMART model (Ghazvininejad et al., 2020b) which also focuses on the training-inference gap in NAR training; we discuss the key differences between our approach and SMART in Section 3.2. Finally, it is important to note that while significant progress has been made, the majority of work in NAR has focused on maximizing the distilled performance. Despite its

importance, NAR performance on raw data remains significantly below leading AR baselines and is the focus on this work.

# 3 CMLMC: Conditional Masked Language Model with Correction

In this section, we analyze the CMLM architecture and identify several weaknesses in its design, namely, lack of token distinguishability and training-inference mismatch. We then propose the CMLMC model, which addresses these problems leading to significant improvements in accuracy.

## 3.1 Token Distinguishability

To generate accurate translations the model must be able to align the source and translated sentences. This is particularly important in Transformer-based architectures where source sentence is typically encoded as memory, and decoder attends over memory during translation. To infer which part(s) of the memory to attend for a given translated token, the decoder needs to know exactly where the token is in the translated sequence and its relative position to other translated tokens. Transformer attention doesn't preserve positional information so in AR models this is achieved by a combination of causal masked attention, positional encodings (PEs), and sequential inference.

Causal masked attention matrix has a lower-triangular shape so each token only attends over tokens that come before it. PEs (fixed or learned) get added to the input token embeddings at each position, and enable the model to infer the relative and absolute distance between tokens (Yan et al., 2019). Finally, sequential inference translates one token at a time conditioning on all previously translated tokens, so the model can identify exactly which token is being translated at each step. Jointly these design choices lead to adequate separation between tokens and provide strong positional information.

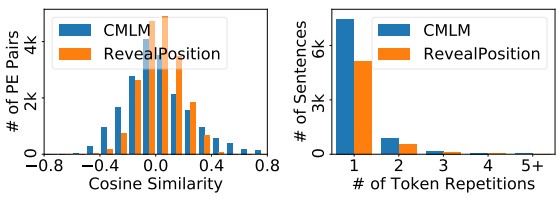

Figure 1: IWSLT'14 De-En 1(a) Binned cosine similarity between all unique pairs of the first 200 well-trained PEs. 1(b) Distribution of token repetitions across test sentences after initial translation from fully masked sentence. For each repetition count, we find the number of test sentences where at least one token consecutively repeats that many times.

In NAR models, parallel translation significantly complicates sentence alignment. In particular, CMLM replaces causal masked attention with a full attention over all tokens. Unmasked attention combined with parallel inference over all tokens implies that the model has to rely solely on PEs for position information. If learned PEs for different positions are very similar to each other, CMLM would not be able to distinguish them. Token indistinguishability is especially problematic during the first inference step since CMLM inference always starts with a fully masked sentence. During this step, all tokens are replaced with the same mask embedding <M> summed into PEs. The model thus has to translate the entire sentence using only PEs to distinguish the input tokens.

Figure 1(a) shows binned cosine similarity between unique pairs of trained PE encodings learned by the CMLM model on IWSLT'14 De-En dataset. We see that the similarity distribution has a long tail on the positive side. Significant number of PE pairs have cosine similarity above 0.5, and over 200 pairs have similarity greater than 0.7. These results indicate that during inference with a fully masked sentence, some token pairs can be nearly indistinguishable for CMLM. This can lead to severe translation errors, the most direct of which is token repetition where indistinguishable tokens get translated to the same word. Figure 1(b) shows the distribution of consecutive token repetitions across test sentences on the same dataset. We see that CMLM can produce over 5 repetitions in a translated sentence, and hundreds of sentences have 3 or more repetitions. It is evident that CMLM has difficulty translating from fully masked sentence, and bad initial translation can be difficult to correct even with multiple steps of iterative refinement.

We address the problem of token distinguishability with a dual strategy. First, each decoder block in CMLM is augmented with a causal masked attention layer, inserted between the unmasked self attention and encoder attention layers, leading to the following block structure: FULL-ATT → MASKED-ATT → ENCODER-ATT → FFN. Re-introducing masked attention brings back the left-to-right sequential hierarchy and breaks symmetry between positions with similar PEs. We

deliberately insert this layer before the encoder attention, as it aligns source and translated sentences, so properly separating input tokens is particularly important there. We use the standard masked attention layer where the upper triangular portion of the SoftMax matrix is set to 0.

Second, we modify how token embeddings and PEs are combined. The majority of transformer-based models simply sum the two embeddings. However, this can be insufficient to propagate the positional information to the upper layers since the model has to balance the scales of both embeddings. This can again be particularly problematic during the fully masked inference step when learned mask embedding <M> is added to PEs. <M> is shared across all masked positions, and in training receives gradient updates that are typically larger in magnitude than individual PEs. Empirically we found that gradient magnitude for <M> can be up to 50x larger than the average gradient magnitude for first 100 most commonly updated tokens throughout training. This can in turn make the magnitude of <M> much larger than PE, making it difficult for the model to distinguish PEs at different positions once they are summed with <M>. To deal with this problem we instead propose to combine token embeddings and PEs with a feed-forward layer (FFN). Formally, given an input token embedding $y_i$ and positional encoding $pe_i$ at position $i$, we first expand by concatenating them and then shrink back down with an FFN: $y_i' = \text{FFN}([y_i, pe_i])$. Here, $[\cdot, \cdot]$ is the concatenation operation and FFN $: \mathbb{R}^{2d} \rightarrow \mathbb{R}^d$ where $d$ is the original embedding dimension. FFN enables the model to appropriately adjust the embedding scales and (de-)emphasize specific dimensions.

The joint effect of these architecture modifications is shown in Figure 1. We refer to this method as RevealPosition. From Figure 1(a) we see that PE cosine similarity distribution is now mainly clustered around 0, and no pair has similarity above 0.5. This is a significant improvement from CMLM where hundreds of PE pairs have cosine similarity above 0.7. Furthermore, from Figure 1(b) we also see that RevealPosition consistently reduces the number of sentences that have repetitions across all repetition counts. In particular, the most frequent one and two token repetitions are reduced by over 30% and 35% respectively. In the experiments section we further show that these relatively simple modifications lead to considerable improvements in BLEU of over 1 point. However, despite the reduction in repetitions, RevealPosition can still make mistakes, so in the next section we proposed a new loss function that aims to teach the model how to correct them.

## 3.2 TRAINING/INFERENCE MISMATCH

It is generally accepted that training and inference procedures should match as closely as possible (Ranzato et al., 2015; Mihaylova & Martins, 2019). When this is the case, improving training loss during optimization typically translates to better inference performance. In AR models, training and inference are well aligned, in both cases the model translates one token at a time conditioned on all previously translated tokens. The only major difference is that during training previously translated tokens are fixed to ground truth, but during inference they are set to model predictions.

This is not the case in many NAR models, including CMLM. The training strategy in CMLM follows a masked language model approach similar to BERT (Devlin et al., 2019) pretraining: a random subset of tokens from the target sentence $Y$ get masked, effectively splitting $Y$ into masked tokens $Y_{mask}$ and observed ground truth tokens $Y_{obs}$; masked tokens are replaced with the mask embedding <M>, and the sentence is passed through the decoder to get predictions for $Y_{mask}$. The loss function then aims to maximize the probability of masked tokens for reconstruction accuracy. Unlike the fixed 15% masking typically used in BERT, CMLM uses a more aggressive strategy, sampling the number of masked tokens uniformly between 1 and sentence length. During inference CMLM always starts with a fully masked sentence and translates all tokens. Iterative refinement are then applied, where a subset of tokens with the lowest probability is re-masked and re-translated at every iteration.

Comparing these training and inference procedures we can see a mismatch. During training CMLM is not trained to correct its predictions. In particular, it is not trained to recognize the errors made in the crucial first inference step. As we discussed in the previous section, translations from this first step can have significant errors (repeated tokens etc.), so learning to correct these errors can be critical to model performance. To better align training and inference, we build an error correction mechanism into our model by adding a correction loss term during training. This term focuses on correcting mistakes after inference with fully masked input. Formally, given a source sentence $X$ and a target sentence $Y$, we first split $Y$ into $Y_{obs}$ and $Y_{mask}$ as in CMLM training. We then apply the decoder to the fully masked sentence $Y_{\varnothing}$, where every token is replaced with <M>, obtaining

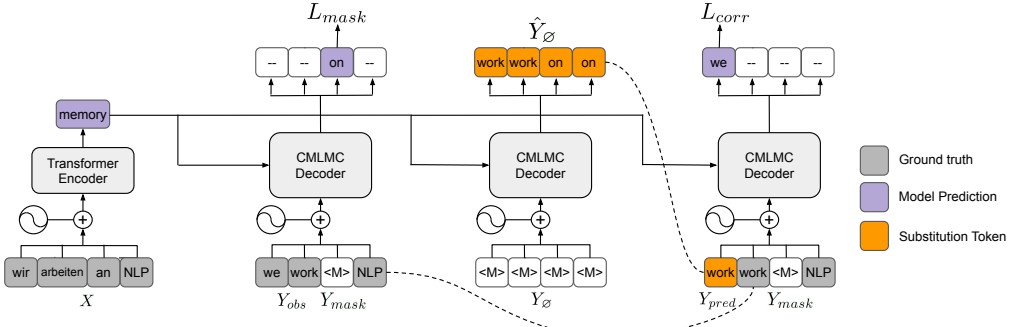

Figure 2: CMLMC loss example. Here, source German sentence $X$=[wir arbeiten an NLP] is translated to the target English sentence $Y$=[we work on NLP]. First, sampled mask $Y_{mask}$ masks out the [on] token. Masked sentence is passed through the CMLMC decoder to predict the masked out token in the $L_{mask}$ loss. Then, fully masked sentence $Y_\varnothing$ is passed through the CMLMC decoder that translates all tokens simultaneously to $\hat{Y}_\varnothing$=[work work on on]. Sampled first token [work] from $\hat{Y}_\varnothing$ is substituted for [we], and the resulting sentence [work work <M> NLP] passes through the CMLMC decoder to correct the [work] $\rightarrow$ [we] in the $L_{corr}$ loss.

$\hat{Y}_\varnothing = \text{Decoder}(X, Y_\varnothing)$. Here, $\hat{Y}_\varnothing$ is identical to the output of the first inference step, where all tokens are predicted at once and can contain significant errors. With probability $p$ we replace each token $y \in Y_{obs}$ with a corresponding predicted token $\hat{y} \in \hat{Y}_\varnothing$. This splits $Y_{obs}$ into two sets: $Y_{pred}$, where tokens are substituted with predictions from $\hat{Y}_\varnothing$, and the remaining portion $Y_{obs} \backslash Y_{pred}$. Our correction loss then aims to predict the tokens in $Y_{pred}$:

$$L_{corr} = - \sum_{y \in Y_{pred}} \log(P(y | Y_{pred}, Y_{obs} \backslash Y_{pred}, Y_{mask}, X)) \tag{1}$$

To compute this loss we make a forward pass through the decoder with a sentence that now contains mask $Y_{mask}$, predictions $Y_{pred}$, and unmasked ground truth tokens $Y_{obs} \backslash Y_{pred}$. The model generates predictions for every substituted token in $Y_{pred}$, and we maximize the probability of the corresponding ground truth tokens. This process approximates the correction procedure, where model re-translates a subset of unmasked tokens by conditioning on current translation. Early self-correction steps primarily condition on tokens from $\hat{Y}_\varnothing$ obtained during the initial pass. As shown in (Ghazvininejad et al., 2019), these early steps are very important and lead to large improvements in BLEU. So the correction loss $L_{corr}$ aims to gradually optimize the model for this process during training. In addition to $L_{corr}$, we keep the original mask prediction loss from CMLM:

$$L_{mask} = - \sum_{y \in Y_{mask}} \log(P(y | Y_{obs}, Y_{mask}, X)) \tag{2}$$

Note that $Y_{pred}$ is dropped from this loss and conditioning is done on ground truth tokens in $Y_{obs}$. The final loss that we use in our model is a combination of correction and mask prediction losses:

$$L_{CMLMC} = L_{corr} + L_{mask} \tag{3}$$

Figure 2 illustrates how the joint loss is computed for an example German to English translation. The mismatch between NAR training and inference procedures is also recognized by SMART (Ghazvininejad et al., 2020b). Similarly to our approach, SMART applies the decoder during training to generate predictions for a subset of tokens and then learns to self-correct these predictions while simultaneously reconstructing masked tokens. Our approach however has two key differences. First, SMART generates token predictions from a partially masked sentence $\{Y_{mask}, Y_{obs}\}$. While this does teach the model to self-correct, the decoder always sees some ground truth tokens $Y_{obs}$ as input. This procedure thus does not address the key first inference step where tokens are predicted from the fully masked sentence with no ground truth input. We discussed that the most significant mistakes are made during this step (also empirically shown in CMLM) so learning to correct them is particularly important. We consequently argue that our approach to instead apply the decoder to fully masked sentence $Y_\varnothing$ for self-correction loss better addresses the training/inference gap. Second, SMART uses the same input with masked tokens $Y_{mask}$ and decoder predicted tokens $Y_{pred}$ for both mask reconstruction and self-correction tasks. This can potentially hamper training since to predict masked tokens decoder also conditions on $Y_{pred}$ which can contain significant errors especially early in optimization. To preserve clean signal in masked training we only condition on ground truth tokens $Y_{obs}$ in $L_{mask}$. Empirically, we demonstrate that our approach leads to significant improvements in accuracy particularly on undistilled data.

## 3.3 CMLMC TRAINING AND INFERENCE

In previous sections we introduced modifications to CMLM that improve token distinguishability and better align training with inference. Combined they form our CMLMC approach. Algorithm 1 outlines the training procedure for CMLMC. Given a training corpus $S = \{(X_i, Y_i)\}_i$ of source and target sentence pairs, for each pair $(X, Y)$ we first sample mask $Y_{mask}$ uniformly from the interval $[1, |Y|]$. We then compute correction and mask losses $L_{corr}$ and $L_{mask}$ as outlined in Section 3.2, and update model parameters using gradients. During inference we follow the same iterative approach used in CMLM. First, a fully masked sentence $Y_{\varnothing}$ is passed through the decoder that translates all tokens at once to get $\hat{Y}_{\varnothing}^{(1)}$. In subsequent iterations $t \geq 1$, a subset of tokens with the lowest likelihood in $\hat{Y}_{\varnothing}^{(t)}$ is masked and re-translated by making a pass through the decoder to get $\hat{Y}_{\varnothing}^{(t+1)}$. The length of

---

**Algorithm 1:** CMLMC Training

**input**: paired training data $S = \{(X_i, Y_i)\}_i$,
  learning rate $\eta$, token substitute probability $p$
initialize model parameters $\Theta$
**while** *not convergent* **do**
  **for** $(X, Y) \in S$ **do**
    sample mask: $Y \to Y_{mask}$ and $Y_{obs}$
    make forward pass:
      $\hat{Y}_{\varnothing} = \text{Decoder}(X, Y_{\varnothing})$
    substitute tokens:
      $Y_{obs} \to Y_{pred}$ and $Y_{obs} \backslash Y_{pred}$
    compute loss:
      $L_{CMLMC}(X, Y) = L_{corr} + L_{mask}$
    update model:
      $\Theta = \Theta - \eta \frac{\partial}{\partial \Theta} L_{CMLMC}(X, Y)$
  **end**
**end**
**return** $\Theta$

---

$Y_{\varnothing}$ is predicted by the special LENGTH token that is appended to the source sentence in the encoder (Ghazvininejad et al., 2019). We repeat the inference procedure for top-$k$ predicted lengths and select the translation with the highest average token probability, analogous to using beam search in AR models.

## 4 EXPERIMENTS

We evaluate our approach on multiple public NMT datasets: IWSLT'14 De-En/En-De, WMT'14 De-En/En-De, and WMT'16 Ro-En/En-Ro. We use the same training/validation/test sets as in previous work (Ghazvininejad et al., 2019) and report test set performance in BLEU for direct comparison. For each dataset we compute performance on both raw and distilled settings, resulting in 12 dataset in total. Distillation is done using the Transformer$_{small/base/large}$ AR model (Vaswani et al., 2017) for the IWSLT'14/WMT'16/WMT'14 datasets respectively, accounting for model capacity and dataset size. We compare CMLMC against leading NAR baselines including, Flowseq (Ma et al., 2019), CMLM and its variants, CMLM+SMART and CMLM+RawPrior (Ghazvininejad et al., 2019; 2020b; Ding et al., 2021), LevenshteinNAR (Gu et al., 2019), DisCo (Kasai et al., 2020), ENGINE (Tu et al., 2020), ReorderNAT (Ran et al., 2021), GLAT (Qian et al., 2021), as well as the models using the alignment losses. Descriptions for all baselines are in the related work section.

All experiments are done using the Fairseq library (Gehring et al., 2017). To stay consistent with previous work, on the IWSLT'14 dataset we use the Transformer$_{small}$ configuration 512-1024-4, while on the WMT datasets we use the Transformer$_{base}$ configuration 512-2048-8 for encoder and decoder in CMLMC. The numbers correspond to embedding dimension, FFN layer size, and number of attention heads respectively. Hyper-parameters for each dataset are selected through grid search and are listed in Table B.1 in Appendix. We apply our modifications to CMLM using the code released by the authors[1]. In all experiments we use a linear FFN to combine the token and PE embeddings. For datasets where CMLM performance is not reported, we train the model ourselves using the original code, and choose the hyper-parameters by applying the same grid search as in CMLMC. We also implement SMART to match the performance on the reported datasets since code is not publicly available, this implementation is then used to evaluate SMART on datasets not reported in the original paper. For CMLMC training we compute the correction loss $L_{corr}$ by randomly substituting 30% ($p = 0.3$) of tokens in $Y_{obs}$ with predictions from $\hat{Y}_{\varnothing}$; the substitution percentage is chosen by grid search from [0.1, 0.2, 0.3, 0.5]. At inference time we follow the procedure outlined in Section 3.3. As in CMLM, we use 10 iterative refinement steps, but lower the beam search length to 3. Adam optimizer (Kingma & Ba, 2015) with default settings is used for all experiments, and we train the models on the IBM servers with 160 POWER9 CPUs, 600GB RAM and 4 Tesla V100 GPUs (32G).

---

[1]https://github.com/facebookresearch/Mask-Predict

Table 1: Results and ablation study on the WMT datasets. * Indicates our training results, as the original papers did not report results on these datasets. For CMLM we report re-run results from (Kasai et al., 2021) as they are better than those reported in the original paper.

| | **Model** | **Param** | **WMT'14 De-En** | | **WMT'14 En-De** | | **WMT'16 Ro-En** | | **WMT'16 En-Ro** | |
|---|---|---|---|---|---|---|---|---|---|---|
| | | # | raw | distill | raw | distill | raw | distill | raw | distill |
| | **AR Transformer** (Kasai et al., 2020; 2021) | 65M | 31.09 | 31.8 | 27.74 | 28.3 | 34.46 | 34.8 | 34.16 | 34.6 |
| NAR | **Flowseq** (Ma et al., 2019) | 258M | 28.29 | 30.68 | 23.64 | 25.31 | 32.91 | 32.84 | 32.35 | 32.20 |
| NAR | **CMLM** (Ghazvininejad et al., 2019) | 67M | 29.40* | 31.20 | 24.61 | 27.40 | 32.87* | 33.31 | 32.86 | 33.7 |
| NAR | **CMLM+SMART** (Ghazvininejad et al., 2020b) | 67M | 29.58* | 31.27 | 25.10* | 27.65 | 32.86* | 33.53* | 32.71* | 33.85* |
| NAR | **CMLM+RawPrior** (Ding et al., 2021) | 67M | – | – | – | 27.8 | – | 33.7 | – | – |
| NAR | **LevenshteinNAR** (Gu et al., 2019) | 67M | – | – | – | 27.73 | – | 33.02 | – | – |
| NAR | **DisCo** (Kasai et al., 2020) | 67M | – | 31.31 | 25.64 | 27.34 | 32.25 | 33.25 | – | 33.22 |
| NAR | **ENGINE** (Tu et al., 2020) | 67M | – | – | – | – | – | 34.04 | – | – |
| NAR | **ReorderNAT** (Ran et al., 2021) | 46M | – | 31.13 | – | 26.49 | – | 31.99 | – | 31.70 |
| NAR | **GLAT** (Qian et al., 2021) | 73M | – | 31.02 | – | 26.55 | – | 33.84 | – | 32.87 |
| Aligned Loss | **CMLM+AXE** (Ghazvininejad et al., 2020a) | 67M | 24.90 | 27.90 | 20.40 | 23.53 | 31.42 | 31.54 | 30.47 | 30.75 |
| Aligned Loss | **CMLM+OAXE** (Du et al., 2021) | 67M | 26.8 | 30.2 | 22.4 | 26.1 | – | 33.3 | – | 32.4 |
| Aligned Loss | **Imputer** (Saharia et al., 2020) | 67M | – | **31.8** | 25.0 | 28.2 | – | 34.1 | – | 34.4 |
| Aligned Loss | **FullyNAT** (Gu & Kong, 2021) | ≈89M | – | 31.39 | 23.58 | 27.49 | – | **34.16** | – | 33.79 |
| Ours | **CMLM+RevPos** | 73M | 30.02 | 30.84 | 25.53 | 27.70 | 33.69 | 33.92 | 33.41 | 34.23 |
| Ours | **CMLM+Corr** | 67M | 30.55 | 31.31 | 26.10 | 28.19 | 33.98 | 34.08 | 33.75 | 34.31 |
| Ours | **CMLMC** | 73M | **30.92** | **31.41** | **26.40** | **28.37** | **34.13** | **34.13** | **34.14** | **34.57** |
| Ours | **CMLMC$_{456}$** | 63M | 30.59 | 31.23 | 26.31 | 27.91 | 33.94 | 33.93 | 33.86 | 34.47 |

### 4.1 RESULTS

Results on the WMT datasets are shown in Table 1. We see that CMLMC outperforms all NAR benchmarks using CE loss, in many cases by a wide margin. On the distilled setting, with the exception of WMT'14 De-En, CMLMC improves over CMLM by more than 0.8 BLEU points, and achieves new state-of-the-art results for Cross-Entropy-based NAR translation on all datasets. Even when compared to models using alignment loss, CMLMC only falls short on WMT'14 De-En when compared to Imputer (Saharia et al., 2020), but is still superior to all other baselines despite using the standard CE loss. Similarly, on the raw setting CMLMC improves over CMLM by 1.5 to over 2 BLEU points on all datasets. Improvements of this magnitude lead to CMLMC largely closing the gap to AR performance on undistilled data. On all datasets except En-De the difference between raw CMLMC and AR is less than 0.5 BLEU which to the best of our knowledge is the first time when such small performance difference is achieved without distillation. This implies that it is possible to train NAR models without distillation and achieve comparable translation quality to AR counterparts. So we believe that with further research in this area, the performance difference between the two frameworks can be eliminated. Jointly, these results, and in particular the improvement over CMLM, indicate that the identified problems do cause a performance bottleneck. They also indicate that our proposed solutions are effective and can lead to significant improvements, while being conceptually simple and easy to implement. Results on the IWSLT'14 datasets are shown in Table A.1 in the Appendix. CMLMC also shows considerable improvements of over 0.7 BLEU on distilled datasets and over 1.8 BLEU on raw datasets compared to the leading baselines.

Analysis of iterative refinement is shown in Table 3(a). For both CMLM and CMLMC we compute BLEU accuracy at different iterations. We also compute total inference time to translate the entire IWSLT'14 De-En test set and fraction of token repetitions. Fraction of repetitions is computed by counting all consecutively repeating tokens and dividing by the total number of translated tokens. This number can thus be interpreted as the repetition rate that we expect to see from each model, for example at 5% we expect to see 50 repetitions in every 1000 translated tokens. From Table 3(a) we see that the repetition rate is consistently lower in CMLMC. This can partially explain the over 3 points gain in BLEU after the initial translation from fully masked sentences (iteration 1). This gain is preserved throughout the refinement iterations, and after the last iteration CMLMC has repetition rate that is almost 50% lower than CMLM. The architectural changes to distinguish tokens together with correction training, enable CMLMC to start with fewer mistakes and more effectively correct them during iterative refinement. We also see that additional masked attention layers in the decoder blocks add an overhead during inference forward passes. However, after the full 10 iterations this overhead is around 20% which we do not believe to be significant for a model of this size. Contrasting CMLMC with SMART, we note that CMLMC outperforms SMART on all datasets, with an average BLEU gain of 0.54 in the distilled datasets and 1.15 in raw, further supporting our arguments in section 3.2.

| Iter | CMLM | | | CMLMC | | |
|---|---|---|---|---|---|---|
| | BLEU | Time | Repetition | BLEU | Time | Repetition |
| 1 | 26.52 | 15.0s | 7.42% | 29.77 | 16.9s | 5.47% |
| 2 | 28.61 | 18.5s | 4.05% | 32.20 | 21.2s | 2.47% |
| 5 | 30.56 | 29.6s | 1.58% | 33.79 | 35.2s | 0.78% |
| 7 | 31.22 | 38.5s | 1.18% | 33.94 | 46.7s | 0.53% |
| 10 | 31.80 | 52.2s | 0.83% | 34.21 | 63.5s | 0.35% |

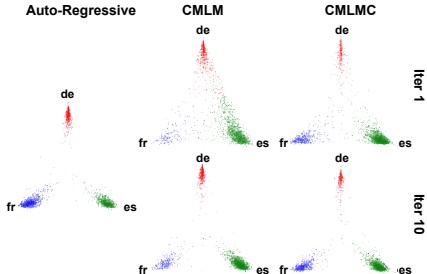

Figure 3: 3(a) IWSLT'14 De-En BLEUs, inference times, and token repetitions for different number of correction iterations. Inference time measures the wall time from when model and test data are loaded until the last sentence has been translated. 3(b) Translation language probability visualization for a multi-target En-De/Es/Fr corpus. Each models is trained to translate all three languages.

## 4.2 ABLATION STUDY

Ablation results on the WMT datasets are shown in Table 1; IWSLT'14 ablations are shown in Table A.1 in the Appendix. Here, CMLM+RevPos incorporates the architectural changes in the decoder to better distinguish tokens (Section 3.1) but is trained with the original CMLM loss. On the other hand, CMLM+Corr keeps the original CMLM architecture but incorporates the correction loss during training (Section 3.2). We observe that both CMLM+RevPos and CMLM+Corr improve performance over CMLM on all datasets in both raw and distilled settings. The improvements are significant and range from 0.2 to over 2 points gain in BLEU. CMLM+Corr generally leads to larger improvement, demonstrating the importance of training the model to correct mistakes. Additional ablation study on the effect of Masked Attention layers and concatenation of PEs are shown in Table A.1, indicating the necessity of both in the RevealPosition mechanism. The gains over CMLM are more pronounced on raw setting than distilled. We hypothesize that a smaller impact on distilled setting is due to the simplification of the underlying training data structure. Knowledge distillation compresses multiple modes in the raw data into a single mode learned by the AR model (Zhou et al., 2020). NAR models trained with this data are less likely to make translation mistakes related to multi-modality so correction is not as important in this case. We still see however, that even on distilled data CMLM+Corr produces significant improvements of up to 1.3 BLEU points. Finally, combining both approaches in CMLMC gives additional improvement, and in all cases CMLMC outperforms both CMLM+RevPos and CMLM+Corr. This indicates that distinguishing tokens and error correction are complementary and jointly lead to better translations.

Architectural modifications to the decoder outlined in Section 3.1 introduce additional parameters through weights in masked attention layers and input FFN. From Table 1 we see that CMLMC has around 9% more parameters than CMLM. To remove the performance effect of extra parameters, we also train a smaller model CMLMC$_{456}$ where input embedding dimension is reduced from 512 to 456; this model has 63M parameters which is 6% smaller than CMLM. As seen in Table 1, the reduction in parameters does impact the performance, however, CMLMC$_{456}$ still outperforms nearly all baselines especially on raw data. This demonstrates that most performance gains in CMLMC come from our modifications and not from additional parameters.

## 4.3 MULTI-MODALITY

We discussed that knowledge distillation is a time-consuming and non-standard step, yet all leading NAR models rely on it since their raw scores significantly underperform the distilled counterparts. Previous work (Zhou et al., 2020) argues that distillation primarily helps to resolve the multi-modality problem in the raw dataset, which NAR models cannot handle directly. Following that work, we use the aligned sentences from the Europarl corpus to create a multi-target English to German, Spanish and French (En-De/Es/Fr) corpus. For every En source sentence, we have three target sentences in De, Es, and Fr, effectively creating a tri-modal data. For each architecture we train a single model on raw data to translate in all three languages by sharing the encoder/decoder layers but learning token representations for each language. During inference the model is applied to En test sentences, and we estimate the probability of translation belonging to each of the three target languages by computing relative count of translated tokens from each language. Figure 3(b) visualizes the results where probabilities close to 1 are shown at the extremes of the corresponding languages.

Table 2: Qualitative examples of test sentence translation from CMLM and CMLMC on the IWSLT'14 De-En dataset. For both models we show translations after the first self-correction iteration (from fully masked sentence) and after the last iteration.

| | |
|---|---|
| **Source:** | also, werde ich sie ihnen zeigen. das ist einer, zwei, drei, vier, fünf. |
| **CMLM iter 1** | so, i'm going to show you. this is **one one**, two, **three three**, four, five. |
| **CMLM iter 10** | so, i'm going to show you. this is one, **two, two**, three, four, five. |
| **CMLMC iter 1** | so, i'm going to show **you you**. this's one, two, three, four, five. |
| **CMLMC iter 10** | so, i'm going to show you. that's one, two, three, four, five. |
| **Source:** | das entspricht der leistung von hundert atomkraftwerken, weil das geht besonders schnell überall. |
| **CMLM iter 1** | that's equivalent of a hundred **nuclear nuclear power power power power**, because it's all going very quickly. |
| **CMLM iter 10** | that's the equivalent of a hundred of nuclear **power power power**, because it's going on very quickly. |
| **CMLMC iter 1** | that's the performance of a hundred **nuclear nuclear power power**, because it's going fast. |
| **CMLMC iter 10** | that's the performance of a hundred of nuclear power plants, because it's going very fast. |

We see that at iteration 1 CMLM outputs a highly mixed distribution where translations have tokens from multiple languages, which clearly shows the multi-modality problem. After 10 iterations of self-correction CMLM is able to separate the languages and mostly produces translations where the majority of tokens are from one target language. This demonstrates that iterative refinement is an important component of NAR translation that helps the models settle into one mode. We also see that CMLMC has a much better language separation at iteration 1 than CMLM. So our proposed modifications improve the multi-modality problem even when translation is done from fully masked sentence. By comparing the graphs from CMLM and CMLMC at iteration 10 vs autoregressive baseline, we observe that they all exhibit a similar degree of separation. However, as we have seen from other experiments their BLEU scores are typically quite different. This suggests that the multi-modality problem might not be the only issue NAR models face when trained on raw datasets. We believe that an additional investigation is necessary here and leave it for future work.

### 4.4 QUALITATIVE ANALYSIS

IWSLT'14 De-En translation examples are shown in Table 2. We focus on the common NAR problem of token repetition and show two example sentences that are representative of the repetition errors that CMLM typically makes. We also show the final translation after 10 iterative refinements and compare against the CMLMC translation. The first sentence has a sequence of numbers "one, two, three, four, five". Embeddings for number tokens tend to be close together in the learned space, and CMLM has difficulty removing these repetitions. After the first iteration "one" and "three" tokens repeat, and while CMLM is able to correct these, it can't fully remove repetitions and the final translation still has repeating "two" token. CMLMC on the other hand has no difficulty with the number tokens and translates them correctly after the first iteration.

The second sentence shows a long repetition that is difficult to correct. Here, "power" is repeated four times by CMLM after the first iteration. After 10 iterations CMLM is able to correct the "nuclear" repetition and remove one "power" token but the other three remain. Since each refinement iteration re-translates a subset of tokens with the lowest probability, longer repetitions are increasingly harder to correct as all repeating tokens need to eventually end up in the lowest probability set. We see that CMLMC also repeats "power", but only once, allowing the correction mechanism to come into play and fix the error. These examples demonstrate that improved token distinguishability can lead to initial translations with fewer repetition mistakes. Our model is then the able to more effectively correct them, resulting in the better end translations.

## 5 CONCLUSION

We introduce the **C**onditional **M**asked **L**anguage **M**odel with **C**orrection (CMLMC), an NAR translation model that addresses the design shortcomings in leading NAR approaches. Through a dual strategy of revealing positional information and adding error correction mechanism, we significantly improve NAR translation performance on both raw and distilled datasets. In particular, when trained on raw data, CMLMC approaches the performances of leading AR models which to the best of our knowledge is the first such result in NAR. Future work involves expanding our approach to other NAR models and further investigation into the relationship between multi-modality and distillation.

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

# A CMLMC Performance on IWSLT'14 En-De Dataset

Table A.1: Results and ablation study on the IWSLT'14 De-En and En-De datasets. * Indicates CMLM and SMART models that were trained by us, as (Ghazvininejad et al., 2019) and (Ghazvininejad et al., 2020b) does not report results on these datasets.

| Model | Param # | IWSLT'14 | | | |
| --- | --- | --- | --- | --- | --- |
| | | De-En | | En-De | |
| | | raw | distill | raw | distill |
| **AR Transformer**(Liu et al., 2020) | 38M | 34.66 | 35.30 | 28.56 | 29.26 |
| **NAR-Reg** (Wang et al., 2019) | 46M | – | 28.04 | – | – |
| **NAR-Hint** (Li et al., 2019) | 46M | – | 28.80 | – | – |
| **Flowseq** (Ma et al., 2019) | 73M | 24.75 | 27.55 | – | – |
| **CMLM** (Ghazvininejad et al., 2019) | 38M | 31.80* | 33.42* | 25.60* | 27.59* |
| **CMLM+SMART** (Ghazvininejad et al., 2020b) | 38M | 30.74* | 33.48* | 24.55* | 27.74* |
| **ENGINE** (Tu et al., 2020) | 67M | – | 33.17 | – | – |
| **CMLM+MaskedAttn** | 44M | 33.08 | 33.70 | 26.03 | 27.93 |
| **CMLM+ConcatPE** | 39M | 33.23 | 33.90 | 26.12 | 27.90 |
| **CMLM+RevPos** | 46M | 33.50 | 33.96 | 26.38 | 28.13 |
| **CMLM+Corr** | 38M | 33.90 | 34.53 | 26.92 | 28.39 |
| **CMLMC** | 46M | **34.28** | **34.78** | **27.55** | **28.51** |
| **CMLMC**$_{456}$ | 38M | 33.83 | 34.44 | 27.35 | 28.47 |

# B CMLMC Hyperparameters

Table B.1: CMLMC hyper-parameters.

| Parameters | IWSLT'14 | WMT'14 | WMT'16 |
| --- | --- | --- | --- |
| learning rate | 0.0005 | 0.0007 | 0.0005 |
| warmup | 30k | 40k | 15k |
| dropout | 0.3 | 0.2 | 0.3 |
| updates | 175k | 150k | 120k |
| epochs | 300 | 250 | 200 |
| GPU | 1xTesla V100 | 4xTesla V100 | 4xTesla V100 |
| tokens/GPU | 8192 | 8192 | 8192 |

