# OpenReview forum: "Improving Non-Autoregressive Translation Models Without Distillation"
_ICLR.cc/2022/Conference — ICLR 2022 Poster_

### Official Review · Reviewer_6uRn · 2021-11-02

**Correctness:** 2
**Technical Novelty And Significance:** 2
**Empirical Novelty And Significance:** 2
**Recommendation:** 3
**Confidence:** 4

**Main Review:**

#### Strengths:
1. Training a NAR model without the help of the autoregressive model is appealing and important.
2. The performance improvements are impressive, which remarkably narrows the performance gap to the autoregressive model without the knowledge distillation.

#### Weaknesses:
1. The novelty is somewhat limited. The indistinguishability of decoder inputs and the mismatch problems have been discussed in previous works[1][2], the authors do not present a clear discussion among them.
2. To my understanding, the correction loss works very similarly to SMART[2]. It is unclear to me the key differences between them and why the former works better than the latter, though the author has discussed it on Page 5.
3.  Although this paper focuses on the performances trained with the raw datasets, several comparisons may not be fair for distilled datasets. Some results of baselines in Table 2 use the Transformer-base as a teacher in WMT14 instead of Transformer-large.

#### Questions:
1. Is the distinguishability of the decoder inputs matter to an iterative-based model? Figure 1 only analyzes the first inference step but misses the 4- or 10-iterations results.  I know this issue is important to fully non-autoregressive models and discussed by Wang et al. 2019[1]. Need to say, their method can be incorporated directly into your setting.
2. I notice that the CMLMC model achieves significant improvements in the first inference step than the CMLM model (Figure 3, it is also very curious to report a Table in the Figure environment). What is the result of WMT14? One-step results can be directly compared with most NAR models, which will provide a stronger baseline for subsequent work.

#### Typos:
1. Which "GLAT (Haviv et al., 2021a)"?

#### References
1. Wang et al. AAAI-2019, Non-Autoregressive Machine Translation with Auxiliary Regularization
2. Ghazvininejad et al. 2020. Semi-Autoregressive Training Improves Mask-Predict Decoding.
3. Qian et al. ACL-2021. Glancing Transformer for Non-autoregressive Neural Machine Translation.

**Summary Of The Paper:**

This paper argues that the indistinguishability of tokens and the mismatch between training and inference leads to inferior translation quality in the previous state-of-art non-autoregressive~(NAR) model --- CMLM.
To address the above issues, the authors propose the CMLMC model and achieve better performances without the help of the autoregressive model.
Specifically, CMLMC first re-introduces a causal masked attention layer to the CMLM model and uses a feed-forward layer to combine the tokens and positional encodings to enhance the distinguishability of the decoder inputs.
Then, the CMLMC model introduces a correction loss that predicts the correct tokens based on the predicted tokens,  avoiding the mismatch between the training and inference stages.
Experiments results on several translation benchmarks show the CMLMC model achieves impressive performance improvements.

**Summary Of The Review:**

Overall, although the paper presented impressive results, it missed some meaningful discussions, such as the importance of indistinguishability and why its correction loss works better than SMART. In addition, it needs to polish the draft.

---

> ### Author Response · Authors · 2021-11-16
> **Response to Reviewer 6uRn (1/2)**
>
> We thank the reviewer for taking the time to evaluate our work and provide valuable feedback. Here, we address the main concerns that were brought up.
>
> 1. CMLMC novelty compared to NAT-REG [1]: We disagree with the reviewer here, and believe that [1] did not investigate the indistinguishability of decoder ***inputs***. NAT-REG is a modified version of NAT [3], and inherits some of its characteristics including generating decoder input with a uniform mapping from encoder input embeddings. This implies that NAT-REG should not have a problem with distinguishing adjacent decoder input tokens, as they are linear mappings of different encoder input embeddings and would mostly be different by construction. NAT-REG's auxiliary loss terms focuses on the repeating/similar tokens in decoder ***output***, forcing them to be different from adjacent tokens with a penalty, as well as preserving enough information to re-construct the source sentence in a back translation. As the reviewer suggested, it would be interesting to combine NAT-REG loss terms with CMLMC to see whether it further reduces token repetition, but we believe that it is a different approach and does not affect the novelty of our submission.
>
> 2. Difference between CMLMC Correction and SMART [2]: Please refer to our comments to Reviewer dfXP on the impact of increasing masking percentage in SMART and how it's different from Correction in CMLMC. In addition, [4] has shown that CMLM's one-step inference performance persists even when up to 75\% of the decoder inputs are masked, as long as the remaining 25\% are set to ground truth tokens. This supports our claim that, when inferencing with Mask-Predict, the first few inference steps conditioned on mostly masked inputs are key to NAR performance. It also implies that CMLMC's way of generating model output from fully masked sentence presents a much more challenging learning task compared to SMART, which in expectation has 50\% of ground truth tokens revealed in the sentence.
>
> [1] Non-Autoregressive Machine Translation with Auxiliary Regularization, Wang et al. AAAI'19
>
> [2] Semi-Autoregressive Training Improves Mask-Predict Decoding, Ghazvininejad et al. 2020
>
> [3] Non-Autoregressive Neural Machine Translation, Gu et al. ICLR'18
>
> [4] Hybrid-Regressive Neural Machine Translation, Wang et al. 2020

---

> > ### Author Response · Authors · 2021-11-16
> > **Response to Reviewer 6uRn (2/2)**
> >
> > 3. Benchmark model configurations: Our work builds on the CMLM approach which uses Transformer large as the distillation model for the WMT14 En-De dataset. To properly benchmark against CMLM we followed the released code and used the same teacher for distillation. We do agree with the reviewer that this can provide a slight advantage over baselines that use the Transformer base teacher, and will add these results in the revised version. We ran preliminary experiments on the WMT14 En-De and De-En datasets distilled with Transformer base teacher, and observed only a small drop in CMLMC performance, 28.11 vs 28.37 on En-De and 31.32 vs 31.41 on De-En. However, the main point of our work is to show that NAR models can be successfully trained on raw datasets and achieve competitive accuracy. We believe that the raw data performance of CMLMC demonstrates that this is indeed possible.
> >
> > 4. Distinguishability with iterative refinement: We believe that the distinguishability of decoder inputs does matter in an iterative refinement NAR model, as shown by the performance gap between CMLM and CMLM+RevPos. As we discussed, the main issue is the first inference step where input tokens are identical except for the learned PEs. This can leads to significant errors including multiple repeated tokens with high model confidence. It is possible that with enough refinement iterations these errors would get corrected. But this would make inference prohibitively expensive and eliminate the speed advantage on NARs. We agree that it would be interesting to incorporate NAT-REG's auxiliary losses with CMLMC, and leave this investigation to future work.
> >
> > 5. CMLMC 1-step performance: We evaluated performance for CMLM and CMLMC on 1'st and 10'th refinement iteration across all datasets:
> >
> > | Model | WMT14 DeEn raw | WMT14 DeEn distill | WMT14 EnDe raw | WMT14 EnDe distill | WMT16 RoEn raw | WMT16 RoEn distill | WMT16 EnRo raw | WMT16 EnRo distill |
> > | ----------- | ----------- | ----------- | ----------- | ----------- | ----------- | ----------- | ----------- | ----------- |
> > | CMLM 1 iter | 23.65 | 28.96 | 18.76 | 25.01 | 30.07 | 32.38 | 29.50 | 32.43 |
> > | CMLM 10 iter | 29.40 | 31.20 | 24.61 | 27.40 | 32.87 | 33.31 | 32.86 | 33.70 |
> > | CMLMC 1 iter | 25.43 | 29.66 | 20.56 | 26.34 | 30.69 | 33.20 | 30.88 | 33.25 |
> > | CMLMC 10 iter | 30.92 | 31.41 | 26.40 | 28.37 | 34.13 | 34.13 | 34.14 | 34.57 |
> >
> > We see that CMLMC reaches competitive performance even with a single refinement step, outperforming CMLM on all datasets. Note that single step inference is not the design intention for either CMLM or CMLMC and the Correction mechanism plays a minimal role in this case. We'd like to emphasize, that our focus in this work is to explore why sequence-level distillation had been the standard practice in NAR model training despite its drawbacks, and whether it can be eliminated.
> >
> > We hope we have addresses all the questions, and would appreciate if the reviewer would re-consider the position on our submission. As stated above, we believe that CMLMC represents a solid step towards learning robust NAR models on raw data. Please let us know if there are any additional concerns.

---

### Official Review · Reviewer_BqHZ · 2021-11-02

**Correctness:** 3
**Technical Novelty And Significance:** 4
**Empirical Novelty And Significance:** 3
**Recommendation:** 8
**Confidence:** 4

**Main Review:**

In general, the paper is mainly about tackling a well-scoped problem (token repetition) using an existing architecture (CMLM), which is well-known in the domain of non-autoregressive MT. The modifications by the authors all make sense, arguably there can be a difference in terms of how each factor contributed to the overall performance improvement, and the authors do show ablated results with a leave-one-out approach. The performance gained by the newly introduced factors looks effective, and they don't seem to overcomplicate the method. The analysis by the authors shows that the proposed ideas help mitigate the multi-modality problem, although the problem is not fully eliminated.

Since the token repetition problem is not fully removed, but reduced to around half the occurrence rate, could the authors confirm that there are still repeated tokens in the translation results?

Overall, I think the effort to remove distillation using AR models is a good direction in general. Eventually, non-autoregressive models will become useful by removing this constraint.

**Summary Of The Paper:**

This paper investigates a few different ways to mitigate token repetition issues in non-autoregressive machine translation (MT) systems, which is also known as the multi-modality problem. The authors highlight that this is mainly due to the mismatch between training and inference in non-autoregressive (NAR) models. In addition to the mismatch, the indistinguishability of tokens makes the problem worse. The authors explain that autoregressive (AR) models suffer less from the same issues because of a combination of multiple factors, such as causal attention, auto-regressive modelling, and introducing positional embedding (the last point is more specific for the Transformer architecture). The authors introduce an alternative path for combining the positional embedding with token embedding using a shallow neural network (RevealPosition), introduce intermediate causal attention between self-attention and cross-attention, and finally a correction loss to deal with repeated tokens.

**Summary Of The Review:**

This paper shows interesting observation points to understand why the multi-modality problem occurs in non-autoregressive MT systems, and proposes a few ways to mitigate the issue. The modifications seem effective and improve both raw and distilled performances. More impressively, the gains are bigger in the raw setting.

I also find that it is nice to remove the restriction of non-autoregressive models that rely heavily on autoregressive models.

---

> ### Author Response · Authors · 2021-11-16
> **Response to Reviewer BqHZ**
>
> We thank the reviewer for taking the time to evaluate our work and provide valuable feedback. Here, we address the main concerns that were brought up.
>
> Token repetition: We do still observe some repeated tokens in the translated sentences from CMLMC. We agree with the reviewer that the improvements introduced in CMLMC (RevealPosition and Correction mechanism) do not completely resolve the multi-modality/token-repetition problem, but alleviate it in a simple and effective manner. Recent work ([1], [2]) had shown that alignment losses are more suitable for NAR training in place of standard cross-entropy, and can mitigate token repetition. It will be interesting to see whether such losses can further improve CMLMC on resolving the multi-modality problem as they can be directly incorporated, we leave such investigation to future work.
> We hope that we have addressed all the questions; please let us know if there are any additional concerns.
>
> [1]: Aligned Cross Entropy for Non-Autoregressive Machine Translation, Ghazvininejad et al, ICML'20
>
> [2]: Order-agnostic Cross Entropy for Non-Autoregressive Machine Translation, Du et al, ICML'21

---

### Official Review · Reviewer_dfXP · 2021-11-03

**Correctness:** 3
**Technical Novelty And Significance:** 3
**Empirical Novelty And Significance:** Not applicable
**Recommendation:** 8
**Confidence:** 4

**Main Review:**

**Strengths**

* The paper is very clearly written and easy to follow.
* The issues behind performance gap between NAR and AR models have been clearly explained and methods to circumvent those issues have been well motivated and justified.
* The resulting model CMLMC achieves state of the art scores for semi-autoregressive machine translation, and more importantly achieves better performance on non-distilled datasets.

**Questions/Concerns**
* *More Ablation Experiments*: In case it is easy enough to execute, it will be informative to see results for `SMART + RevPos`  in Table 1 as well.


* *Comparison with SMART (I)* : In Section 3.2, the authors discuss the similarities and key differences with a similar approach to bridge train-test mismatch for NAR models - SMART (Ghazvininejad et al. 2020). The authors say -  "First, SMART generates token predictions from a partially masked sentence $\{Y_{mask}, Y_{obs}\}$. While this does teach the model to self-correct, the decoder always sees some ground truth tokens $Y_{obs}$ as input." This is slightly wrong, as in SMART, the masking length can be set to sequence length which would then result in predictions from a fully masked sequence (1st inference step). CMLMC's approach can be considered as a re-weighted form of SMART, where all weight is given to all masked sequence (which the authors demonstrate performs better empirically). if the authors agree with this, it would be great to clarify this in the paper.


* *Comparison with SMART (II)*: Second key difference, as the authors mention, lies in the introduction of an additional correction loss instead of modifying the existing CE loss function. While the arguments presented by the authors make intuitive sense, it will be nice to perform some empirical assessment of this claim. i.e. comparing CMLMC with re-weighted SMART.


* Question: Did the authors try adjusting relative weights of correction loss and CE loss?


**Summary Of The Paper:**

Authors describe two important issues behind performance gap between semi-autoregressive CMLM (conditional masked language models) and autoregressive models in Machine Translation - (i) indistinguishability of tokens, and (ii) training and inference mismatch. They propose Conditional Masked Language Model with Correction (CMLMC) to address these issues through intermediate causal attention layers, and FFN layers for combining token and positional encodings for (i) and adding a correction loss for the first inference step for (ii). Authors show that CMLMC outperforms existing NAR methods on many benchmarks when trained on distilled and non-distilled datasets.

**Summary Of The Review:**

The paper is well written, presents well motivated improvements for improving NAR models. The empirical evidence is convincing, and proper ablation experiments have been presented. I have provided some suggestions for experiments and technical corrections to the authors. I believe this paper presents interesting findings which will be relevant to the machine translation community. I recommend acceptance for the paper.

---

> ### Author Response · Authors · 2021-11-16
> **Response to Reviewer dfXP**
>
> We thank the reviewer for taking the time to evaluate our work and provide valuable feedback. Here, we address the main concerns that were brought up.
>
> 1. SMART+RevPos: Due to time and resource constraints, we ran SMART+RevPos on 6 WMT datasets, and present the performances below together with the BLEU scores for SMART and CMLMC for easy comparison; trainings on the WMT14 De-En datasets are in progress.
>
> | Model      | WMT'14 En-De raw |  WMT'14 En-De distill |  WMT'16 Ro-En raw |  WMT'16 Ro-En distill |  WMT'16 En-Ro raw |  WMT'16 En-Ro distill |
> | ----------- | ----------- | ----------- | ----------- | ----------- | ----------- | ----------- |
> | SMART | 25.10 | 27.65 | 32.86 | 33.53 | 32.71 | 33.85 |
> | SMART+RevPos | 25.26 | 27.76 | 33.35 | 33.37 | 32.89 | 34.11 |
> | CMLMC | 26.40 | 28.37 | 34.13 | 34.13 | 34.14 | 34.57 |
>
> We see that RevPos improves SMART performance on multiple datasets, demonstrating the robustness of our approach. However, SMART+RevPos is still inferior to CMLMC on all datasets, further demonstrating the effectiveness of the Correction mechanism in CMLMC.
>
> 2. Correction mechanism vs. SMART with sequence length masking: while we agree that it is possible to set the decoder input masking length to sequence length, doing so significantly degrades model performance for both SMART and CMLM. We ran CMLM+SMART+FullMask on both raw and distilled IWSLT14 De-En datasets and observed a 2.6 BLEU drop on the distilled dataset, and over 8 BLEU drop on the raw dataset. We attribute this degradation to the fact that CMLM needs to learn dictionary embeddings from scratch, and aggressive masking can hamper that by not showing sufficient ground truth tokens to guide learning. Similar findings were shown for BERT-like pre-training, where masking more than 15\% of tokens leads to drop in performance [1]. A key improvement in CMLMC is that we separated the fully masked decoder pass from the traditional CMLM training step by using the additional $L_{corr}$ loss. This allows us to both correct mistakes made from fully masked input, as well as provide sufficient information to learn accurate dictionary embeddings. Therefore, we believe that the Correction mechanism in CMLMC is different and novel compared to SMART even with sequence length masking.
>
> 3. Relative weights between losses: we have not explicitly experimented with adjusting the relative weights between $L_{corr}$ and $L_{mask}$. We agree that it would be an interesting experiment to run and thank the reviewer for bringing it up. As mentioned in Section 4, we did experiment with changing the substitution token percentage when training the Correction mechanism (30\% achieves the best results) which can be viewed as indirectly adjusting the weight on the $L_{corr}$ loss.
>
> We hope that we have addresses all the questions; please let us know if there are any additional concerns.
>
> [1] Exploring the Limits of Transfer Learning with a Unified Text-to-Text Transformer, Raffel et al, JMLR'20

---

### Official Review · Reviewer_ssed · 2021-11-03

**Correctness:** 4
**Technical Novelty And Significance:** 2
**Empirical Novelty And Significance:** 3
**Recommendation:** 8
**Confidence:** 4

**Main Review:**

**Strengths**
- The proposed method is very simple, and can be applied to many iterative NAR models (and perhaps more, e.g., when we use BERT-like models for generation in general).
- The experiments show the strong performance compared to prior NAR models, especially when distillation is not applied.
- Removing the necessity of distillation will have a practical impact for NAR models because distillation is, as authors point out, time and resource consuming.

**Weaknesses**
- While it is correctly acknowledged in Sec. 4.1, the gap between raw and distillation setting for WMT14 EN-DE is very substantial. This might be because of the diverging and relatively flexible word order of German, which makes the multimodality problem more severe. This suggests that the proposed method has a clear limitation in addressing the multimodality.
- (Question) Since the modified objective involves two separate decoder passes for every batch (for Lmask and Lcorr), I suspect that translates to an increase in training time. How does this increase compare to the training time increase you would get from knowledge distillation? If the proposed method's increase is much smaller than applying knowledge distillation, it would mean that the method can save training overhead compared to approaches that require knowledge distillation.

**Updates**
My question above is fully addressed in the response. I would strongly recommend that the authors add the discussion on increased training time to the final version, as one major benefit of removing KD is training time savings. I increased the score from 6 to 8.

**Summary Of The Paper:**

This paper presents two techniques that improve iterative non-autoregressive machine translation (NAR). The first technique modifies the conditional masked language modeling objective from prior work in a way that makes training more similar to iterative inference. The other improvement is the way positional information and tokens are encoded. This is designed to encourage the model to distinguish between target positions and avoid repetitions.

**Summary Of The Review:**

Overall, this paper presents s simple, effective approach to improve NAR machine translation. Their claim is supported by their experiments. I still have some concern described above.

---

> ### Author Response · Authors · 2021-11-16
> **Response to Reviewer ssed**
>
> We thank the reviewer for taking the time to evaluate our work and provide valuable feedback. Here, we address the main concerns that were brought up.
>
> 1. WMT14 En-De raw/distill performance gap: we agree that the performance gap can be due to the flexible word order for German (En-De translation is harder than the other direction) which adds additional modalities and makes the problem more challenging. While CMLMC doesn't completely resolve the multi-modality issue, it is a simple and effective method of considerably reducing this problem, as shown by our superior performance on the raw data compared to all previous NAR models.
>
> 2. Training time: we thank the review for bringing up this point. The increase in training time for CMLMC is typically around 60\% of CMLM's training time, which costs less than training an AR teacher for distillation. For example, on the IWSLT14 En-De dataset the per epoch training time for a 512-1024-4 model is: 175s for AR transformer; 91s for CMLM; 153s for CMLMC, which is less than the previous two combined. Moreover, to achieve the best performance typically a larger AR model is used for distillation (for example Transformer large on the WMT14 En-De dataset) which can take significantly longer to train. Beyond increased time and added complexity, distillation can also remove important information from the training leading to problems such as lexical choice error on low frequency words [1]. For these and other reasons, we believe that removing distillation is an important research direction in NAR.
>
> We hope that we have addressed all the questions, and would appreciate if the reviewer would consider increasing the score; please let us know if there are any additional concerns.
>
> [1] Understanding and Improving Lexical Choice in Non-Autoregressive Translation. Ding et al, ICLR'21

---

### Public Comment · ~Fei_Huang3 · 2021-11-10
**A Question**

Great work, I find the paper is very interesting. May I ask a question?

Previous work[1] shows that CMLM with 10 iterations are actually slower than a deeper-encoder shallow-decoder autoregressive model (and  also worse in translation quality). I am curious about the quality-speed trade-off of your method against the deeper-encoder shallow-decoder transformer. (Does the Wall Time in Figure 3 use batch size=1?)

Moreover, I find that Table 1 mixes iterative NATs and non-iterative NATs. It would be great to specify the number of iterations (or speed) in Table 1.

[1] Deep Encoder, Shallow Decoder: Reevaluating Non-autoregressive Machine Translation. ICLR2021

---

### Decision · Program_Chairs · 2022-01-20

**Decision:**

Accept (Poster)

**Comment:**

This paper proposes a modification of the training objective of non-autoregressive MT which claims most of the improvements that other approaches obtain only through knowledge distillation (KD) from an autoregressive teacher.

The strategy has been largely appreciated as simple and the results suggest that it's rather effective. One of us was not ready to accept certain aspects of the comparisons in the paper, and challenged the paper also from a speed of generation point of view. While I see that NAT-MT is very much concerned with speed, removing the dependency on an autoregressive teacher is an important step in the NAT-MT agenda (as KD has various drawbacks, e.g., it corrupts the statistics of the training data), and, in my view, disentangling the two desiderata (e.g., faster models, and no KD) is okay at this stage. I hope the authors will not take this recommendation as a reason to ignore the comments in that review, rather, that they take it as an opportunity to address those comments as well as possible (for example, by positioning the work more carefully wrt speed and the possibly negative impact of KD).

On style: Table 1 should fit within the margins of the paper, please fix it. Also, avoid boldfacing model names and avoid vertical bars (please check this nice guide on making [tables look nice](https://people.inf.ethz.ch/markusp/teaching/guides/guide-tables.pdf)).